# Activities of H₂O₂-Converting Enzymes in Apple Leaf Buds during Dormancy Release in Consideration of the Ratio Change between Bud Scales and Physiologically Active Tissues

Anna M. Hubmann *, Alexandra Jammer and Stephan Monschein *

Institute of Biology, Plant Sciences, University of Graz, Schubertstraße 51, 8010 Graz, Austria
* Correspondence: anna.hubmann@edu.uni-graz.at (A.M.H.); stephan.monschein@uni-graz.at (S.M.)

**Abstract:** Hydrogen peroxide-converting enzyme activities in leaf buds of the apple cultivar Idared during the transition from dormancy release to the ontogenetic development were investigated. For this purpose, leaf buds were collected from 26 March 2021 (DOY = day of the year 85) to 23 April 2021 (DOY 113) and the air temperature was continuously monitored. Enzyme assay protocols for catalase (CAT), intracellular peroxidase (POX), and cell wall-bound peroxidase (cwPOX) in apple leaf buds were successfully established based on published protocols. All enzymes showed considerable changes in activity during the observation period. Fluctuation in daily mean air temperatures seemed not to affect the activities of POX and CAT, whereas severe drops in daily mean air temperature may have interrupted the assumed trajectory of cwPOX activity during the stage of ontogenetic development. In addition, the importance of considering changes in the ratio between physiologically active tissues and bud scales when investigating physiological changes in buds during the phase of dormancy release and ontogenetic development is discussed. A new reference system, namely the "adjusted dry weight" [aDW], is proposed to circumvent this shift in ratios when working with scaled buds.

**Keywords:** antioxidative enzyme activities; catalase; peroxidase; cell wall-bound peroxidase; bud dormancy; dormancy release; *Malus × domestica* BORKH.; apple leaf buds





## 1. Introduction

Oxygen (O₂) is a prominent electron acceptor in aerobic organisms [1]. Since molecular O₂ is unable to accept two electrons at a time, reactive oxygen species (ROS) are generated as by-products in plant aerobic metabolism [2]. The oxygen-containing reactive molecules exist in ionic states such as hydroxylic radicals ($^{\bullet}OH$) and superoxide anions ($O_2^{\bullet-}$), or molecular states, including hydrogen peroxide ($H_2O_2$), and singlet oxygen ($^1O_2$) [3]. ROS are continuously produced in plant cells. In particular, metabolic activities with high rates of oxidation or with excessive electron flows, such as respiration and photosynthesis, are major sources of ROS production [4], making ROS ubiquitous in plant cells.

Due to their high reactivity, ROS can cause damage to membranes through peroxidation of lipids, oxidation of proteins, enzyme inhibition, and damage to nucleic acids, and eventually lead to programmed cell death [5]. The counterpart to the cytotoxic properties of ROS is an antioxidative system comprising non-enzymatic antioxidants and enzymatic antioxidants. These act synergistically to maintain the delicate equilibrium of ROS generation and scavenging in plant cells. The state when ROS generation exceeds the scavenging capacity of the antioxidative system in plant cells is referred as to "oxidative stress", which ultimately leads to cell death [6]. Various environmental stress factors, such as salinity [7], drought [8], and low temperatures [9], enhance ROS accumulation and perturb scavenging, which triggers oxidative damage [10]. Other than their harmful properties, increasing evidence suggests that ROS act as signal molecules, playing important parts at various developmental and growth stages in plants under normal conditions, as well as in resistance

enhancement in response to stress [11–13]. $H_2O_2$ is especially thought to act as a signal molecule: The moderately reactive oxygen derivate has a relatively long half-life (1 ms) compared to other ROS [14] and a high permeability across membranes [15]. Thus, $H_2O_2$ is thought to act as a second messenger for signals generated by other ROS [14,16]. Generally, at high concentrations, $H_2O_2$ leads to programmed cell death [17]; at low concentrations, however, it can trigger processes in the response to biotic and abiotic stresses, and in plant growth and development [18]. Controlled $H_2O_2$ production in plant cells during plant–pathogen interactions or under various abiotic stress conditions induces resistance enhancement [19]. Other than its participation in resistance mechanisms, $H_2O_2$ has also been suggested to play a role as a signal molecule during normal, "stress-free" growth and development of plants [17].

Since $H_2O_2$ is an omnipresent oxygen derivate in plant cells, a variety of enzymes with various isoforms have evolved to balance the equilibrium of its production and reduction. Among the most important $H_2O_2$-scavenging enzymes are catalases (CAT; EC 1.11.1.7) and peroxidases (POX; EC 1.11.17) [20]. Although the basic function of both enzymes, namely the conversion of $H_2O_2$, is identical, their enzymatic properties and scope of functions vary: catalases do not require an electron donor to convert $H_2O_2$ into $H_2O$ and $O_2$ because $H_2O_2$ is used by CAT as both an acceptor and a donor of hydrogen (Figure 1) [21,22]. In contrast, peroxidases convert $H_2O_2$ using specific molecules as electron donors. Peroxidases generally prefer phenolic compounds as reductants for the conversion of $H_2O_2$ into $H_2O$ (Figure 1) [23].

$$2\ H_2O_2 \xrightarrow{\text{CAT}} 2\ H_2O + O_2$$

$$H_2O_2 \xrightarrow[\underset{\text{(red.)}}{AH_2} \quad \underset{\text{(ox.)}}{A}]{\text{POX}} 2\ H_2O$$

**Figure 1.** Schematic representation of the reactions of hydrogen peroxide ($H_2O_2$)-scavenging enzymes catalase (CAT) and peroxidase (POX). CAT does not require an electron donor for the conversion of $H_2O_2$ into water ($H_2O$) and oxygen ($O_2$). POXs use a range of molecules as reductants ($AH_2$), for instance, phenolic compounds. By the reduction of $H_2O_2$ into $H_2O$, $AH_2$ is oxidized into A [21,24].

The catalytic rate of CAT is high, whereas its affinity to $H_2O_2$ is relatively low [15]. The $K_M$ of CAT for $H_2O_2$ has been estimated to be between 40 mM and 600 mM [25], while the $K_M$ of POX may be in a range of 15–20 mM (with guaiacol as the electron donor) [26]. CAT is thought to be predominately involved in the detoxification of hydrogen peroxide rather than in its regulation as a signal molecule [27]. This assumption is supported by the observation that CAT activity often increases concomitantly with the overall activation of the antioxidative system under adverse environmental conditions [28]. Plant peroxidases exist in both soluble (intracellular peroxidases; POX) and ionically bound forms (cell wall-bound peroxidase; cwPOX) [29]. Numerous studies report that POXs play important roles throughout the plant life cycle, including processes such as cell wall metabolism (through cwPOX), lignification, auxin metabolism, fruit growth and ripening, ROS and reactive nitrogen species metabolism, and pathogen defense [30,31].

Due to their wide range of functions, $H_2O_2$-scavenging enzymes have also aroused interest in the context of dormancy research in perennial fruit trees such as apple trees. Dormancy in perennials is an adaptive process, which is associated with a delay in development and growth [32]. The definition of dormancy according to Lang [32], i.e., "dormancy is the absence of visible growth in any plant structure containing a meristem" is the most common and most widely used. As the suspension of growth and development can be triggered by exogenous and endogenous factors, Lang classified dormancy into three categories: paradormancy, endodormancy (true dormancy), and ecodormancy. Lang's classification is based on whether the period of rest is controlled by physiological signals in the plant, but not in the respective structure (e.g., apical dominance; paradormancy); whether the

central control originates from the dormant structure itself (endodormancy); or whether the delay in growth and development is caused by unfavorable environmental conditions (e.g., cold temperatures in spring; ecodormancy) [33]. The transition from paradormancy to endodormancy initiates with the cessation of growth due to declining temperatures and photoperiods [34–36]. Leaves and flowers, which ensure regrowth and reproduction in spring, are formed in buds in the previous year [33]. Endodormancy is assumed to be released by the fulfillment of the chilling requirement of the respective cultivar, due to the accumulation of chilling units [37]. No visible changes occur during endodormancy and ecodormancy, which complicates an accurate detection of the transition phases from endodormancy release to ecodormancy, and ultimately the stage of ontogenetic development [36,38]. The latter one must start some weeks before the first visible phenological state, i.e., bud swelling can be observed in order to initiate processes of development [38]. Many of these dormancy phases can be associated with all kinds of stresses, including low temperature, dehydration, and respiratory and oxidative stress [39]. Accordingly, evidence concerning the role of the antioxidative system in dormancy control and (dormancy) release has been gathered [34]. For instance, Abassi et al. documented low activities of CAT and POX in dormant apple flower buds and increasing activities during the phenological state of bud swelling [40]. With the onset of bud break, POX-activity decreased again, while CAT-activity remained high during flower development. Also, Wang et al. and Wang and Faust showed changes in the activity of POX and CAT during dormancy release in apple buds (lateral and flower buds) after cold, heat, and special dormancy breaking-chemicals (allyl disulfide or thidiazurone) treatment [41,42].

Previous findings, as mentioned above, indicate that antioxidative enzymes such as CAT and POX may play a key part during dormancy release and at the beginning of ontogenetic development in perennial fruit trees. Therefore, enzyme assays for CAT, POX, and cwPOX in apple leaf buds were established according to Fimognari et al. [43]. In vegetative buds of perennial fruit trees, the delicate parts, i.e., the shoot apical meristem and leaf primordia [44], are protected by bud scales during winter. Bud scales, which are modified leaves [45], experience marginal growth during dormancy release, especially compared to the remaining tissues encapsulated in the scales. Thus, in this study, the term "bud scales" will be used for the brown, lignified, and modified leaves which primarily have a protective purpose, showing little or no growth during dormancy release. For all tissues within the bud scales that will develop into leaves, the term "physiologically active tissues" is used. The changes in the ratio between bud scales and physiologically active tissues during growth resumption were considered and enzyme activities were expressed on an "adjusted dry weight (aDW)" basis, i.e., the dry weight of the physiologically active tissues without the bud scales. Once the assays were established, the activities of the key $H_2O_2$-converting enzymes were investigated in leaf buds of apple trees (cultivar Idared) under natural orchard conditions during the transition from dormancy release to the ontogenetic development. The enzymatic activities in leaf buds were comparatively discussed among each other, and with regard to the air temperatures during the observation period in late March and April.

## 2. Materials and Methods

### 2.1. Plant Material

For the establishment of the enzyme assays in apple leaf buds, as well as for the investigation of the enzymatic activities during dormancy release, leaf buds were sampled from fifteen 14-year-old apple espalier trees of the cultivar "Idared", grafted on Malling 9 rootstocks (*Malus × domestica* BORKH.). The trees were located in a plantation of an experimental station for pomiculture and viticulture, "Versuchstation für Obst- und Weinbau", in Haidegg, Graz (coordinates 47.07987, 15.50271). Per sampling date, the collected buds were combined into one large mixed sample. The sampling took place from 26 March 2021 (DOY [day of the year] 85) to 23 April 2021 (DOY 113) on DOY 85, DOY 92, DOY 97, DOY 100, DOY 108, and DOY 113. Sampled buds were immediately blast-frozen in liquid nitrogen,

followed by freeze-drying for at least 48 h. The samples were ground with a vibrating tube mill (Retsch © MM 400) at maximum vibration frequency for 3 min. Until further use, freeze-dried and ground plant material was stored at −80 °C.

### 2.2. "Adjusted Dry Weight [aDW]"—Determination of the Ratio between Bud Scales and Physiologically Active Tissues

For the determination of the ratio between bud scales and physiologically active tissues, fifty leaf buds were collected separately on each sampling date. The fifty buds were divided into five mixed samples, each comprising ten buds. Bud scales were separated from the remaining tissues under a stereo microscope before freeze-drying. After freeze-drying, the dry weight (DW) of scales and remaining tissues of each mixed sample was recorded, and the percentage of scales and remaining tissues on each sampling date was calculated. To express enzyme activity on an "adjusted DW" [aDW] basis, i.e., the mean dry weight of physiologically active tissues without bud scales, the percentage of physiologically active tissues (on each sampling date) was calculated from the respective initial dry weight of the buds.

### 2.3. Air Temperature

Air temperature [°C] was recorded during the sampling period (from DOY 85 to DOY 113) every hour by a data logger positioned in the middle of the tree row. For each day (DOY, defined as the time between sunrises) the average temperatures [°C], as well as the maximum and minimum values per day (per DOY) were calculated from a final data set of over 700 data points.

### 2.4. Extraction Procedure for Intracellular and Cell Wall-Bound Proteins

Both for the establishment of the enzyme assays for vegetative apple buds, as well as for the investigation of the enzymatic activities in leaf buds (per sampling date) during dormancy release, protein extracts were prepared in triplicate (n = 3).

Protein extraction was performed according to a protocol slightly modified from Fimognari et al. [43]. An amount of 100 mg freeze-dried and ground plant material was weighed out into reaction tubes and 1 mL extraction buffer (40 mM Tris-HCl pH 7.6, 1 mM EDTA, 0.1 mM Phenylmethane sulfonyl fluoride, 1 mM Benzamidine, 24 μM NADP, 14 mM β-Mercaptoethanol) was added to the plant material. To bind phenolic compounds, 50 mg PVPP per mL buffer was added to the buffer the day before extraction, to swell overnight. After homogenization, the samples were incubated in an overhead shaker at 4 °C for 30 min. Subsequently, the samples were centrifuged at 14,000 rpm for 10 min at 4 °C (in case of insufficient pellet formation, the samples were centrifuged longer). The supernatant of each sample (crude extract, containing intracellular, water-soluble proteins) was centrifuged for 15 min at 14,000 rpm and 4 °C to remove the remaining particles, while the pellets were kept on ice. The crude extracts were dialyzed overnight against 20 mM $K_2HPO_4/KH_2PO_4$ pH 7.6 buffer at 4 °C. The remaining pellets were washed three times in $ddH_2O$ before they were re-suspended with 1 mL high salt buffer (40 mM Tris-HCl pH 7.6, 15 mM EDTA, 3 mM $MgCl_2$, 1 M NaCl) and incubated overnight in an overhead shaker at 4 °C. Afterwards, the samples were centrifuged at 14,000 rpm for 10 min at 4 °C (or longer in case of insufficient pellet formation). The supernatant (cell wall extracts, containing proteins that are ionically bound to the cell wall) was centrifuged for 15 min at 14,000 rpm and at 4 °C to remove the remaining particles. The cell wall extracts were dialyzed overnight against 20 mM $K_2HPO_4/KH_2PO_4$ pH 7.4 buffer at 4 °C. The extracts were stored at −20 °C in aliquots until further use.

### 2.5. Kinetic Enzyme Assays Procedures

All kinetic assays were performed in triplicate (for each of the three individual extracts) in flat-bottom 96-well plates (Sarstedt, Nümbrecht, Germany) or UV-transmissive flat-bottom 96-well plates (UV-Star, Greiner Bio One, Kremsmünster, Austria) in a microplate photometer (SpectraMax® ABS Plus, Molecular Devices, San Jose, CA, USA). The total

reaction volume in each well for the assays was 160 μL and the extract volumes used for the assay establishments ranged from 0.01 to 3.5 μL. For control reactions, the respective substrate was omitted. Enzymatic activities were expressed on an adjusted dry weight (aDW) basis, as described above.

### 2.5.1. Catalase Assay

The determination of CAT activity in apple leaf buds was carried out according to Fimognari et al. [39]: Crude extracts were incubated with 100 mM $H_2O_2$ and 0.001% Antifoam 204 in 50 mM $K_2HPO_4/KH_2PO_4$ buffer at pH 7. The decrease in absorbance, representing the conversion of $H_2O_2$ (Figure 2) [46], was recorded at 240 nm for 25 min at 30 °C. For control reactions, $H_2O_2$ was omitted. For the calculation of CAT activity from the decrease in absorbance per second, the negative values were multiplied by −1 to receive positive values and the extinction coefficient 43,600 [L mol$^{-1}$ cm$^{-1}$] was used.

$$H_2O_2 + H_2O_2 \xrightarrow{\textbf{CAT}} 2\ H_2O + O_2$$

hydrogen peroxide          water + oxygen
(240 nm)                   (no UV absorbance)

**Figure 2.** Schematic representation of the photometric determination of catalase activity (CAT). The conversion of $H_2O_2$ catalyzed by CAT is followed by the decrease in absorbance at 240 nm [46]. Figure modified after Aebi [46].

### 2.5.2. Peroxidase Assay

The determination of peroxidase (POX) and cell wall-bound peroxidase (cwPOX) activities in apple leaf buds was carried out according to a protocol slightly modified from Fimognari et al. [43]: Crude extracts (for POX) or cell wall extracts (for cwPOX) were mixed with 0.45 mM $H_2O_2$ and 2 mM guaiacol in 100 mM $K_2HPO_4/KH_2PO_4$ buffer at pH 7. For the control reactions, $H_2O_2$ was omitted. The increase in absorbance, representing the oxidation of guaiacol to tetraguaiacol (Figure 3) [47], was recorded at 450 nm for 25 min at 30 °C. For the calculation of POX activity, the extinction coefficient 25,500 [L mol$^{-1}$ cm$^{-1}$] was used.

guaiacol                   tetraguaiacol
(UV absorbance)            (450 nm)

**Figure 3.** Reaction scheme for the photometric peroxidase (POX) and cell wall-bound peroxidase (cwPOX) activity assay. The aromatic oil guaiacol is used by POX/cwPOX as an electron donor for the reduction of $H_2O_2$. Guaiacol (colorless) is oxidized into tetraguaiacol, which can be detected by an increase in absorbance at 420–470 nm [47]. Figure modified from Lopes et al. [48].

## 3. Results

### 3.1. Establishment of the Enzyme Assays for Vegetative Apple Buds

At the initial stage, the amount of extract [μL] that had to be used to measure enzyme activities in apple buds was investigated. Therefore, each kinetic assay was tested for extract-amount-depended linearity to examine the amounts of extract from which dose-dependent linearity between extract and enzyme activity existed. For this step, the original substrate concentration according to Fimognari et al. [43] was used. Subsequently, each

assay was tested for substrate saturation, to ensure that the substrate was not a limiting factor for the reaction. For this purpose, the assays were performed with at least half of the substrate concentrations as well as with at least a two- to threefold increase in the substrate concentrations given by Fimognari et al. [43].

### 3.1.1. Catalase

The CAT activity assay showed dose-dependent linearity of the decrease in absorbance per second detected at 240 nm. In Figure 4A, the negative values of the recorded absorbance change (decrease of absorbance) were multiplied by $-1$ to receive positive values. The CAT reaction was saturated at the initial substrate concentration of 100 mM $H_2O_2$, and substrate inhibition occurred at concentrations above 150 mM (Figure 4B).

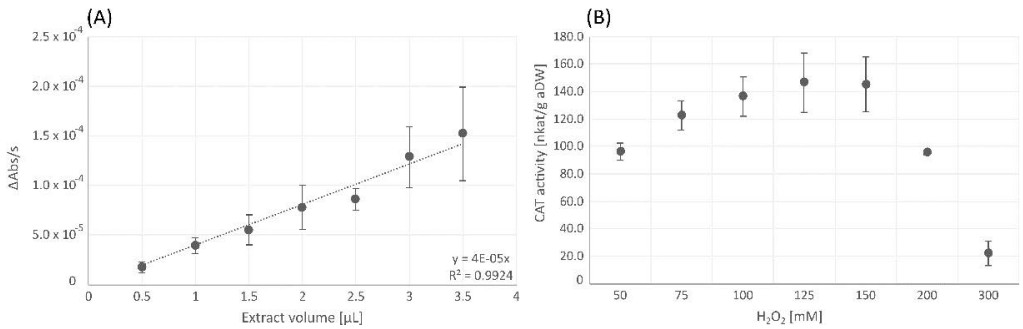

**Figure 4.** Dose-dependent linearity of catalase (CAT) assay, performed with the original substrate concentration of 100 mM according to Fimognari et al. [43] (**A**) and variation of the substrate ($H_2O_2$) concentrations for catalase (CAT) assay (**B**). Changes in absorbance per second [$\Delta$Abs/s] (**A**) and CAT activity [nkat/g aDW] (**B**) are shown as mean values $\pm$ standard deviation (n = 3). The negative values resulting in the decrease of absorbance were multiplied by $-1$ to receive positive values.

### 3.1.2. Peroxidase

The POX activity assay showed dose-dependent linearity of the increase in absorbance per second detected at 450 nm (Figure 5A). Concerning the substrate saturation for POX, the initial substrate concentration of 0.15 mM $H_2O_2$ turned out to be insufficient and had to be increased to 0.45 mM $H_2O_2$ (Figure 5B).

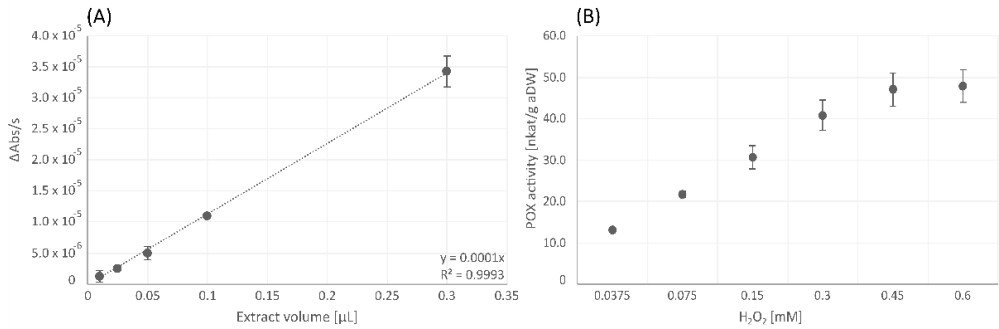

**Figure 5.** Dose-dependent linearity of peroxidase assay, performed with the original substrate concentration of 0.15 mM according to Fimognari et al. [43] (**A**) and variation of substrate ($H_2O_2$) concentrations for peroxidase (POX) assay (**B**). Changes in absorbance per second [$\Delta$Abs/s] (**A**) and peroxidase (POX) activity [nkat/g aDW] (**B**) are shown as mean values $\pm$ standard deviation (n = 3).

### 3.1.3. Cell Wall-Bound Peroxidase

The cwPOX activity assay showed dose-dependent linearity of the increase in absorbance per second detected at 450 nm (Figure 6A). Concerning the substrate saturation for cwPOX, the concentration of $H_2O_2$ had to be increased from the initial 0.15 mM to 0.45 mM to make sure that substrate availability was not a limiting factor in the reaction (Figure 6B).

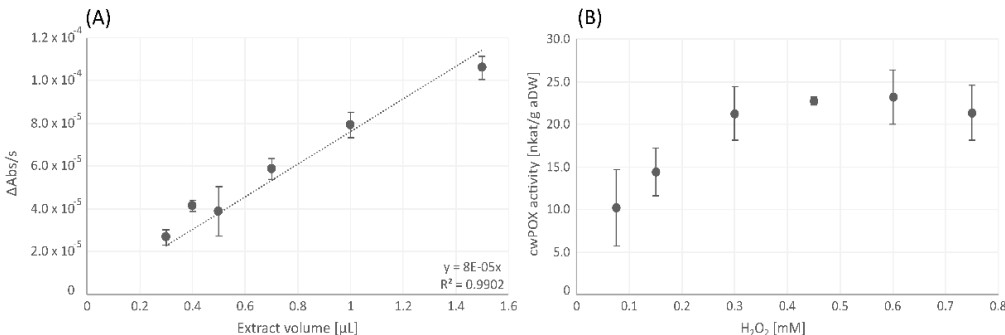

**Figure 6.** Extract-dependent linearity of cell-wall bound peroxidase (cwPOX) assay, performed with the original substrate concentration of 0.15 mM according to Fimognari et al. [43] (**A**) and variation of substrate ($H_2O_2$) concentrations for cell wall-bound peroxidase (cwPOX) assay (**B**). Changes in absorbance per second [$\Delta$Abs/s] (**A**) and cwPOX activity [nkat/g aDW] (**B**) are shown as mean values $\pm$ standard deviation (n = 3).

### 3.2. Investigation of CAT, POX, and cwPOX Activities in Vegetative Apple Leaf Buds of the Cultivar Idared during the Transition from Dormancy Release to the Ontogenetic Development

$H_2O_2$-converting enzyme activities in leaf buds were investigated and subsequently related to air temperatures during the observation period (as seen in the synoptical representation).

#### 3.2.1. Ratio between Scales and Physiologically Active Tissues of Apple Leaf Buds during the Time of Observation

At the beginning of the sampling series (DOY 85), bud scales made up a larger amount (approximately 60%) of the sampled plant material. By DOY 92, the ratio between bud scales and other tissues was relatively similar, with a slightly larger amount of physiologically active tissues (53%). From DOY 92 to DOY 97, the relative amount of physiologically active tissues further increased, so that the proportion between scales and other tissues constantly drifted apart. Between DOY 97 and DOY 100, no considerable changes could be observed. Until the end of the observation period on DOY 113, the amount of physiologically active tissues sharply increased from 61% on DOY 100 to 80% on DOY 113 (Figure 7). Consequently, the aDW of each sampling date was used as the reference system for enzyme activities in apple leaf buds.

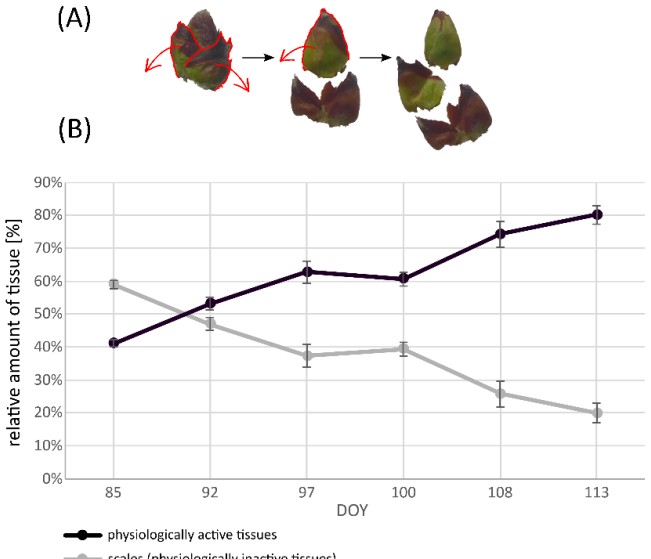

**Figure 7.** Process of descaling apple leaf buds (**A**) and relative amount [%] of bud scales and physiologically active tissues of apple leaf buds on each sampling date, expressed as day of the year [DOY] (**B**). Data points are shown as mean values $\pm$ standard deviation (n = 5).

### 3.2.2. Air Temperature

Air temperature [°C] was continually recorded at one-hour intervals from late March to April. The data logger was positioned in the center of the espalier tree row. Out of the recorded data, a daily mean temperature (per DOY, defined as the time between sunrises), the daily maximum and minimum values (per DOY), as well as the mean temperature intervals before (from DOY 79 to DOY 84) and between sampling dates (intervals from DOY 85 to DOY 91, DOY 92 to DOY 96, DOY 97 to DOY 99, DOY 100 to DOY 107, and DOY 108 to DOY 112) were calculated (Figure 8).

At the beginning of the recording (DOY 79), the daily mean air temperature was below zero. From DOY 79 onwards, until DOY 84 (one day before the first sampling date), daily mean air temperature continuously increased. The mean air temperature during the sampling interval DOY 79 to DOY 84 was 2.1 °C (see Figure 8, orange bar). From DOY 85 to DOY 91 daily mean air temperature further increased. The mean air temperature during the sampling interval DOY 85 to DOY 91 was 11 °C (Figure 8). From DOY 91 onwards, the daily mean air temperature decreased until DOY 96. The mean air temperature during the sampling interval DOY 92 to DOY 96 was 8.8 °C. Afterwards, daily mean air temperatures increased until DOY 100; the mean air temperature during the sampling interval DOY 97 to DOY 99, however, was considerably low at 3 °C (see Figure 8). Between DOY 100 to DOY 107, daily mean air temperature decreased again until DOY 105. From DOY 105 onwards, the daily mean air temperature continuously increased until the end of the observation period. The mean temperature during the sampling interval DOY 100 to DOY 107 was 6.1 °C. The mean temperature during the last sampling interval DOY 108 to DOY 112 was 8.2 °C (see Figure 8).

As shown in Figure 8, the apple trees were protected by sprinkler irrigation during the nights of DOY 97, DOY 98, DOY 104, DOY 105, and DOY 106, due to temperatures below zero. The recorded daily minimum recorded temperatures during these nights ranged from $-5.3$ to $-2.0$ °C.

### 3.2.3. Phenological Growth Stage of Apple Leaf Buds

The developmental stages of leaf buds were photographically documented at the beginning and the end of the observation period. At the beginning of sampling (DOY 85), leaf buds could be categorized as BBCH 00, which means the leaf buds were closed and covered by brown bud scales [49]. At DOY 113, at the end of the observation period, leaf buds were at BBCH 15, meaning that leaves began to unfold, not yet showing the full leaf size [49] (see Figure 8).

### 3.2.4. Hydrogen Peroxide-Degrading Enzyme Activities in Apple Leaf Buds

POX activity was below 30 nkat/g aDW at the beginning of the observation period (DOY 85) and increased to nearly 42 nkat/g aDW at DOY 92, which corresponds to an increase in activity of 40%. From DOY 92 to DOY 97, POX activity increased to approximately 60 nkat/g aDW, corresponding to an increase in POX activity of 50%. From DOY 97 to DOY 100, POX activity showed a further increase, reaching its maximum activity of approximately 90 nkat/g aDW. From DOY 100 onwards, POX activity decreased until the end of the observation period (DOY 113). Generally, the curve progression of POX activity as seen in Figure 8 shows an increase from the beginning (DOY 85) until DOY 100, and a subsequent decrease until the end of the observation period (DOY 113).

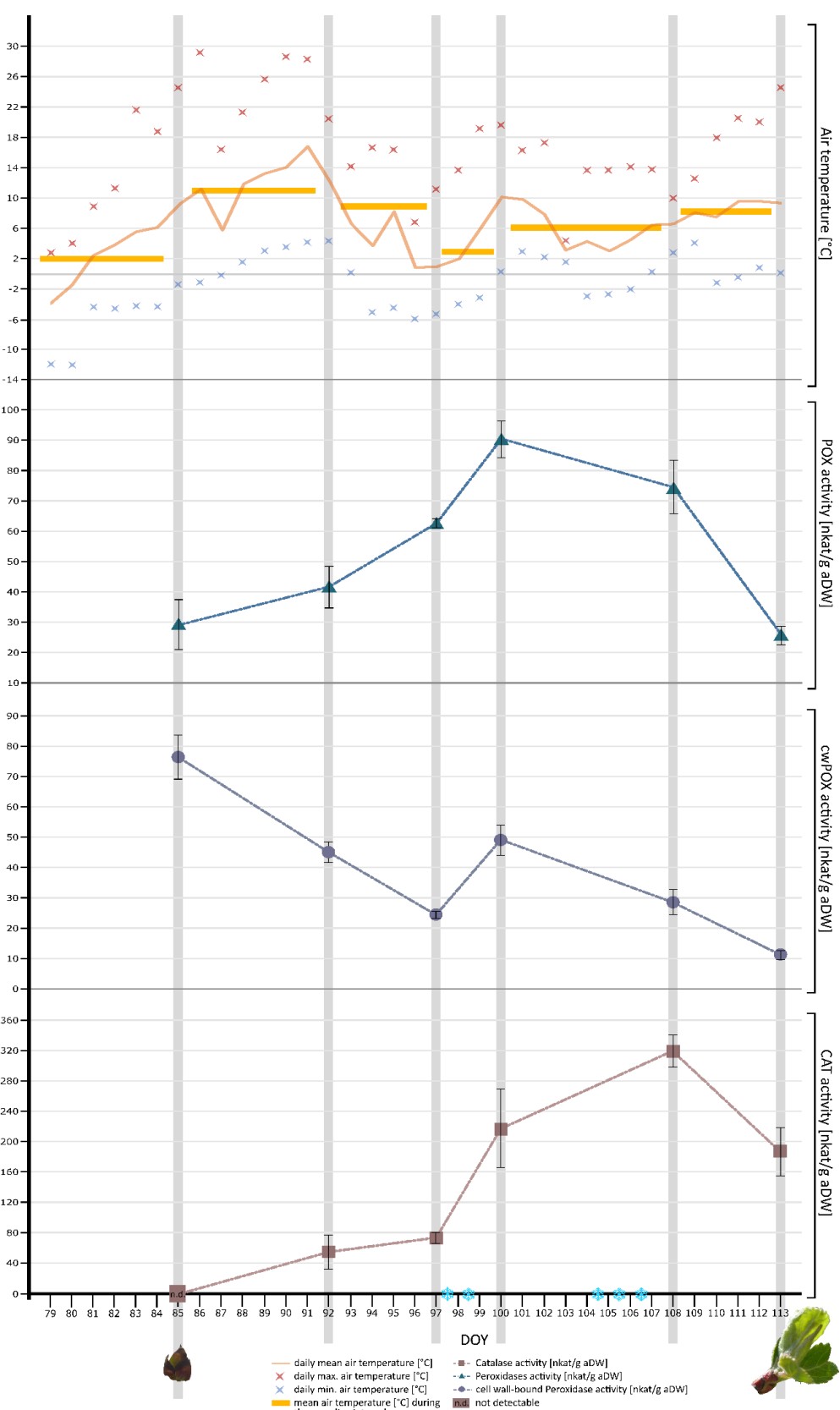

**Figure 8.** Synoptical representation of air temperature and antioxidative enzyme activities (CAT, POX, cwPOX) in leaf buds during the period of observation. The enzymatic activities in nkat/g aDW are shown as mean values ± standard deviation (n = 3). Additionally, the phenological state of leaf buds in the beginning (DOY 85) and at the end (DOY 113) of the observation period are shown.

cwPOX activity was relatively high at the beginning of the investigation period at approximately 76 nkat/g aDW. From DOY 85 to DOY 92, cwPOX activity sharply decreased to approximately 45 nkat/g aDW, which corresponds to a decrease of 70% in cwPOX activity. cwPOX activity then further decreased from DOY 92 to DOY 97 to approximately 24 nkat/g aDW, which corresponds to more than an 80% decrease in cwPOX activity. From DOY 97 to DOY 100, cwPOX activity sharply increased, doubling its activity from approximately 24 nkat/g aDW to 49 nkat/g aDW and then decreased again until the end of the observation period (DOY 113). Generally, the curve progression of cwPOX activity as seen in Figure 8 shows a decrease over the entire observation period with a single peak from DOY 97 to DOY 100.

CAT activity could not be detected at the beginning of the observation period (DOY 85). Subsequently, at DOY 92, CAT activity was around 55 nkat/g aDW and increased to approximately 74 nkat/g aDW at DOY 97, which corresponds to a 35% increase in CAT activity from DOY 92 to DOY 97. CAT activity further increased to approximately 217 nkat/g aDW, nearly tripling its activity from DOY 97 to DOY 100. The highest CAT activity, about 318 nkat/g aDW, was observed on DOY 108. From DOY 108 to the last sampling date (DOY 113), CAT activity decreased. Generally, the curve progression of CAT activity as seen in Figure 8 shows an increase until DOY 108 and a subsequent decrease until the end of the observation period (DOY 113).

## 4. Discussion

To endure low temperatures, freezing, shortened photoperiods, and limited water availability during the winter months, perennial fruit trees linger in a period of rest, referred to as (bud) dormancy. Buds that ensure regrowth and reproduction in spring comprise the shoot apical meristem and the (leaf or flower) primordia, which are protected by bud scales [33,44,50]. The transition from dormancy to active growth, bud break, and, ultimately, blooming represents very sensitive stages in the plant growth cycle: While dormant buds show high tolerance towards adverse environmental conditions such as freezing temperatures [51–54], buds become more and more sensitive during dormancy release [55–57]. Thus, bud break should take place at a time in spring, when conditions are consistently favorable for plant growth and reproduction [58,59]. Not least because of the restricted predictability of climate-changing effects, dormancy release has not only become the subject of intensive research but has also—especially in the case of perennial fruit trees—gained importance for agricultural and horticultural production over the last years.

The fact that no visible changes occur during dormancy complicates an accurate detection and subdivision of its (transition) phases such as endodormancy release, the transition to ecodormancy, and the beginning of ontogenetic development [36,38]. The state of bud swelling, which is associated with a steady rise of (free) water content represents the first visible sign of biological activity [38]. From bud swelling onwards, small milestones of development, which are associated with the accumulation of dry matter [60], become visible and can be divided and registered into phenological growth stages [38]. In order to initiate these developmental processes, the end of ecodormancy and, consequently the beginning of ontogenetic development, must start some weeks before the first observable phenological state of bud swelling [38].

Especially in recent years, physiological and molecular studies on bud dormancy in a broad range of plant species resulted in huge advances in dormancy research and contributed to the current understanding of the mechanisms underlying (early) phenological processes [36,61]. New approaches in dormancy research through non-targeted approaches such as genomics, proteomics, and metabolomics have provided insights into many biochemical processes during transition phases of dormancy that had been thought to be free from cellular activities in previous concepts [36,61–66]. Regarding proteomic methods, for instance, a study on dormant buds of Japanese pear (*Pyrus pyrifolia*, Nakai) revealed that most of the proteins that could be identified were involved in oxidation-reduction processes [67]. Among these proteins, catalase and peroxidases were identified.

The authors concluded that $H_2O_2$, and $H_2O_2$-converting enzymes such as catalase and peroxidase, seem to be involved in the transition from endodormancy to ecodormancy phases [67]. ROS, particularly $H_2O_2$, and sub-lethal oxidative stress have recently been suggested to play a role during dormancy and dormancy release [39,68,69]. Furthermore, Beauvieux et al. [39] proposed ROS as key molecules in cross-linking environmental cues and metabolic processes which regulate plant growth and development.

Excess ROS generation is associated with an upregulation of the antioxidative defense system. ROS-degrading enzymes such as CAT and POX protect plant cells from excessive ROS production [70]. Notably, in many studies which suggest an involvement of ROS and antioxidative enzymes during dormancy release, the sample fruit trees were treated with artificial dormancy-breaking agents, such as hydrogen cyanamide, to promote synchronized bud break [71]. An application of dormancy-breaking chemicals allows more precise statements about when trees overcome dormancy and will start to bloom. However, a comparison of (potted) trees under controlled laboratory conditions with fruit trees under natural orchard conditions (which are exposed to fluctuations in environmental conditions) should be considered with caution. Thus, the scope of this work was to investigate changes in the $H_2O_2$-converting enzyme activities (CAT, POX, and cwPOX) in apple leaf buds of trees of the cultivar Idared during dormancy release under natural conditions in an apple orchard. To discuss physiological changes in leaf buds with an environmental factor that is considered to influence the induction, maintenance, and release of dormancy, and subsequently, the ontogenetic development [44,50,61], air temperature was monitored during March and April.

Since buds enable perennials to continue growth after a period of rest, most studies investigate dormancy-related processes within the buds. However, the bud structure itself represents a major limiting factor in the physiological investigation. For instance, an average freeze-dried apple leaf bud of the cultivar Idared on DOY 85 weighed approximately 4.5 mg (bud scales included), or approximately 2 mg when scales were removed. This indicates that a relatively large number of buds has to be sampled to gather enough material (dry matter) for physiological investigations. Additionally, as this study shows, scales make up a considerable amount of the sampled material. Descaling buds, however, is extremely labor-intensive and time-consuming, particularly with regard to the mentioned number of buds needed. Thus, in many cases, buds are further processed together with scales for more time-efficient handling. Our study proves that, to investigate various substances in buds (which also comprise scales) over a specific period, it is necessary to consider the shift in the relationship between bud scales (which experience little or no growth) and physiologically active tissues encapsulated in the scales, comprising the shoot apical meristem and (leaf) primordia, that experience extensive growth after dormancy release, during ontogenetic development. This is particularly the case if the common reference system "per g dry weight" is used. Because of the progressive morphological development of buds following dormancy release, which is inevitably associated with the accumulation of dry matter [60], the relative amount of physiologically active tissues increases compared to the bud scales. However, at the early developmental stages, bud scales constitute a bigger part of the overall dry weight, as it is shown in Figure 7 where scales made up approximately 60% of bud dry weight on DOY 85. To circumvent this shift in ratios when working with scaled buds, we propose a new reference system, namely the "adjusted dry weight" [aDW] which considers the changing ratios between physiologically active tissues and scales during dormancy and growth resumption. Hence, enzyme activities were expressed on an adjusted dry weight basis, meaning the dry weight of physiologically active tissues the buds comprised on each sampling date.

Regarding the investigation of antioxidative enzyme activities, the respective assays according to Fimognari et al. [43] had to be established for apple leaf buds in the first place. $H_2O_2$-concentrations had to be increased from 0.15 mM to 0.45 mM for POX and cwPOX to ensure substrate saturation (see Figures 5B and 6B). The initial $H_2O_2$-concentration for CAT [43] turned out to be sufficient.

Since ROS and the antioxidative system have been suggested to be involved in the resting phase of perennials, a lot of evidence about these elements playing a role in the control of dormancy and dormancy release has been gathered [34,39]. However, the interpretations of the role of ROS—particularly $H_2O_2$—and, consequently, of $H_2O_2$-converting enzymes during dormancy and its release differ. Many reports, most of them using dormancy-breaking agents, showed increased levels of $H_2O_2$ before bud break. For instance, Pérez et al. [72] determined increasing $H_2O_2$ levels that coincided with an inhibition of CAT in grapevine buds after hydrogen cyanamide application. Also, Pereira et al. [73], who investigated physiological and biochemical changes in persimmon buds, reported an increase in $H_2O_2$ content and antioxidative enzyme activity in dormant buds. During dormancy release, however, $H_2O_2$ levels were elevated, whereas the activities of key enzymes of $H_2O_2$-metabolism were low. Thus, it was suggested that $H_2O_2$ functions as a positive signal in dormancy release [72,74]. On the contrary, according to Porcher et al. [75] the exogenous application of $H_2O_2$ to quiescent axillary rose buds prevented bud outgrowth. Through an enhanced activity of antioxidative enzymes, $H_2O_2$ levels decreased, which allowed buds to resume growth. Also, in grapevine buds, the maximum $H_2O_2$ content was determined when buds were dormant, and a decreasing $H_2O_2$ content could be observed during bud break [69]. These findings suggest that $H_2O_2$ is associated with the depth of dormancy and has a negative regulatory effect on bud break [69,75]. An increased ROS production must be accompanied by a stimulation of the antioxidative defense system to cope with oxidative stress and to prevent lethal cellular ROS levels in plant cells [20]. Consequently, the accumulation of $H_2O_2$ must coincide with increased activities of $H_2O_2$-converting enzymes during dormancy release. Accordingly, observations of Wang et al. [41] and Wang and Faust [42] also indicate that dormancy release coincides with an upregulation of the antioxidative system in apple buds (lateral buds and flower buds) treated with various dormancy-breaking chemicals: in vegetative apple buds, an increase of POX and CAT activities prior to bud break could be shown [41]. Also, Abassi et al. [40] investigated enzyme activities of CAT and POX in apple flower buds. While, initially, both enzymes showed low activities in dormant buds, POX and CAT activities increased during bud swelling. While POX showed a decrease in activity with the onset of bud break, CAT activity continuously remained at a high level throughout flower development [40]. In accordance with the results of these earlier studies, relatively low activities of POX and CAT could be observed in apple leaf buds at the beginning of the sampling period (DOY 85) in the present study (Figure 8). Throughout this timespan, leaf buds were at the BBCH Stage 00 (see Figure 8), meaning the buds were tightly closed and meristematic tissues were protected by bud scales [49]. Thus, it can be assumed that the apple leaf buds were during the transition period between dormancy release and the state of ontogenetic development and subsequently entered the stage of growth resumption during the observation period from March to April. As mentioned above, an elevation of $H_2O_2$ content prior to bud break has been observed in many studies of various perennial fruit crops. Prudencio et al. [76] hypothesized, based on transcriptomic studies on bud dormancy, that the downregulation of peroxidase genes during endodormancy may lead to an accumulation of $H_2O_2$ that modulates a subsequent upregulation of $H_2O_2$-scavenging enzymes, ultimately leading to dormancy release. Also, in grapevine buds, an upregulation of POX was documented during the state of bud swelling after dormancy release [74,77]. Consequently, the increase in POX and CAT activity observed in the present study may serve the purpose of scavenging excess $H_2O_2$ in bud tissues in order to restore conditions that ultimately allow the resumption of growth [76]. Especially the properties of CAT, i.e., a very fast turnover rate [4], an energy-efficient degradation of $H_2O_2$ without requiring cellular reducing equivalents [78], and a comparatively low affinity for $H_2O_2$, suggest that the upregulation of CAT activity might be due to the requirement for the scavenging of excess $H_2O_2$ [79].

While POX and CAT mainly showed increasing activities in apple leaf buds over the whole observation period (from DOY 85 to DOY 100 or DOY 108), cwPOX activity was highest at the beginning of the observation time (DOY 85), during the transition from

dormancy release to the beginning of ontogenetic development and mainly decreased until the end of the observation period, with a single peak in activity from DOY 97 to DOY 100 (Figure 8). Cell expansion during plant growth is associated with the loosening and stiffening of the cell wall [80]. The cessation of growth is related to the stiffening process of the cell wall, associated with the formation of cross-links between cell wall polymers [81–83]. Cross-links to stiffen the cell wall are, inter alia, mediated via cell wall-bound peroxidases, indicating that high cwPOX activities are negatively correlated with plant growth [84–86]. Consequently, when assuming that the apple trees used in our study were at the transition period between dormancy release and the onset of ontogenetic development at the beginning of the sampling period, the decrease in cwPOX activity in leaf buds may be due to the resumption of growth. As mentioned above, air temperature as an environmental factor considered to influence processes during dormancy [44,50,61] was continuously recorded between March and April. From DOY 97 to DOY 99, the sampling interval before cwPOX peaked by almost doubling its activity, the mean air temperature during the sampling interval dropped to 3 °C (see Figure 8). Due to cold stress, including both low and freezing (<0 °C) temperatures, ROS generation is increased, resulting in oxidative stress [87,88]—which could also be observed in the present study—when daily minimum air temperatures dropped below −4 °C during the cold spell between DOY 96 and DOY 99 (see Figure 8). While POX and CAT activities continuously increased during this period (from DOY 97 to DOY 100), the transient peak of cwPOX activity may indicate a modulating effect of low to freezing temperatures on the activity of cwPOX. There are very limited reports about the induction of cwPOX activity due to cold stress. However, He et al. [81] reported an increase of ionically bound cwPOX in banana leaves (of a species with higher cold tolerance) after cold treatment. The authors believe that a fast increase in cwPOX activity is likely to play a key role in the detoxification of $H_2O_2$ in leaves. He et al. argued that, since CAT is predominantly located in peroxisomes, cwPOX represents the main $H_2O_2$-scavenging enzyme in the apoplast of plant cells [89]. After the above-mentioned cold spell between DOY 96 and DOY 100, cwPOX activity decreased again. POX activity continued to decrease from DOY 100 onwards. On the last observation date (on DOY 113), when leaf bud development progressed to the BBCH Stage 15, meaning the leaves began to enfold [49] (see Figure 8), daily mean air temperature was moderate, and the activities of POX and cwPOX were lowest. CAT activity, however, remained relatively high (Figure 8). Similarly, Abassi et al. [40] reported a decrease in POX activity with the onset of bud break and a continuously high CAT activity throughout apple flower development.

## 5. Conclusions

Our work provides a detailed description of a method to determine the activities of three key enzymes of the antioxidative system in a single extract from lyophilized apple leaf buds.

We successfully established enzyme activity assays for the $H_2O_2$-converting enzymes in apple buds according to Fimognari et al. [43] for semi-high throughput determination of antioxidative enzyme activities.

This study provides the following two main findings: Firstly, enzymatic activities of POX, cwPOX, and CAT were investigated between the delicate transition phase of dormancy release and the beginning of ontogenetic development. All three enzymes showed considerable changes in activity during this observation period. Fluctuations in daily mean air temperatures seemed not to affect the activities of POX and CAT, whereas severe drops in daily mean air temperature may have interrupted the assumed trajectory of cwPOX activity during the stage of ontogenetic development. Secondly, when investigating physiological changes in buds during dormancy release, it is of utmost importance to consider the proportions between bud scales and physiologically active tissues. Otherwise, the examined physiological parameters, for instance enzyme activities, may be understated, particularly at the transition period of dormancy release and at the early stages of ontogenetic development. Our present study proposes a way to circumvent this shift in ratios by

using a new reference system, namely the "adjusted dry weight" [aDW], when working with scaled buds.

**Author Contributions:** Conceptualization, S.M., with input from A.M.H. and A.J.; methodology, S.M., A.J. and A.M.H.; investigation, A.M.H.; writing—original draft preparation, A.M.H.; writing—review & editing, A.M.H., A.J. and S.M.; visualization, A.M.H.; supervision, S.M. and A.J.; project administration, S.M. All authors have read and agreed to the published version of the manuscript.

**Funding:** Open Access Funding by the University of Graz.

**Institutional Review Board Statement:** Not applicable.

**Informed Consent Statement:** Not applicable.

**Data Availability Statement:** Not applicable.

**Acknowledgments:** We would like to express our thanks to Leonhard Steinbauer and Thomas Rühmer at the experimental station for pomiculture and viticulture, "Versuchstation für Obst- und Weinbau" in Haidegg (Graz) for providing the sample apple trees for this study.

**Conflicts of Interest:** The authors declare no conflict of interest.

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
