# Peer review of "Activities of H2O2-Converting Enzymes in Apple Leaf Buds during Dormancy Release in Consideration of the Ratio Change between Bud Scales and Physiologically Active Tissues"

_horticulturae, doi:10.3390/horticulturae8110982_

Round 1
Reviewer 1 Report
Results reported in the manuscript are fascinating and the research was usually well performed. Overall this work permits publication in Horticulturae, but I also think authors need to address few concerns in order to take it to next level.
1. Please add some background about apple leaf bud release in the introduction part.
2. Please polish and unify the format of figures in this manuscript.
Reviewer 2 Report
Temporal cessation of flower bud’s growth in autumn is a strategy of perennials to acquire cold /freeze tolerance, as well as to protect the meristematic cells of the developing organs against winter temperatures. In the last few years several physiological studies have proposed the role for different metabolites, enzymes and their activity, as well signaling molecules during winter rest, endo- and ecodormancy, and the ontogenetic development. The meristematic tissues are protected until bud break, leaf unfolding or flowering by (brown) bud scales, and understandable the ratio of growing tissues to the bud scales changes. Molecular biological investigations always require immediate shock freezing after sampling, or isolation, fractionation, in the cold range at -80 degrees Celsius. This is generally impractical. In this article, the relative proportion of bud scales for apple cultivar “Idared” in the period from March to April, after “dormancy release” until “bud break” is shown as an example, which represents a very innovative approach. As a result, the term “adjusted Dry weight” (aDW) is used the enzyme activity of CAT, POX, cwPOX is expressed on this base. After the revision, the MS will make a very interesting and valuable contribution to the extent to which enzymes involved in ROS metabolism react during ontogenetic development.
Comments/remarks/recommendations: see file

Reviewer 3 Report
1. Abstract The part of abstract need to be shorter and the description of the methods shared most of the abstract. It is better to strengthen and point out the main conclusion and results of the study.
2. Sampling:what kind of shoots used for the sampling of leaf bud in the tree? Why you choosed leaf bud not flower bud? Since the apical dominance also may have effect of leaf bud dormancy, did you consider this pont when you sampled the buds?
3. In the part of Result, too much words and Figures in the description of the enzyme analysis, I suggest to reduce the words and figures.
4. The reference format and uniform need to be rewrite according to the guildance of the Journal.
Round 2
Reviewer 2 Report
Congratulations!